# Signaling Role of Mitochondrial Enzymes and Ultrastructure in the Formation of Molecular Mechanisms of Adaptation to Hypoxia

**DOI:** 10.3390/ijms22168636

**Published:** 2021-08-11

**Authors:** Ludmila Lukyanova, Elita Germanova, Natalya Khmil, Lybov Pavlik, Irina Mikheeva, Maria Shigaeva, Galina Mironova

**Affiliations:** 1Institute of General Pathology and Pathophysiology, Baltijskaya Str. 8., 125315 Moscow, Russia; elita.germanova@yandex.ru; 2Institute of Theoretical and Experimental Biophysics RAS, Pushchino, 142290 Moscow, Russia; nat-niig@yandex.ru (N.K.); pavlikl@mail.ru (L.P.); mikheirinal@yandex.ru (I.M.); shigaeva-marija@rambler.ru (M.S.)

**Keywords:** mitochondrial enzymes, catalytic subunits of mitochondrial complexes (MC- I-V), mitochondrial dynamics, adaptation to hypoxia

## Abstract

This study was the first comprehensive investigation of the dependence of mitochondrial enzyme response (catalytic subunits of mitochondrial complexes (MC) I-V, including NDUFV2, SDHA, Cyt b, COX1 and ATP5A) and mitochondrial ultrastructure in the rat cerebral cortex (CC) on the severity and duration of in vivo hypoxic exposures. The role of individual animal’s resistance to hypoxia was also studied. The respiratory chain (RC) was shown to respond to changes in environmental [O_2_] as follows: (a) differential reaction of mitochondrial enzymes, which depends on the severity of the hypoxic exposure and which indicates changes in the content and catalytic properties of mitochondrial enzymes, both during acute and multiple exposures; and (b) ultrastructural changes in mitochondria, which reflect various degrees of mitochondrial energization. Within a specific range of reduced O_2_ concentrations, activation of the MC II is a compensatory response supporting the RC electron transport function. In this process, MC I develops new kinetic properties, and its function recovers in hypoxia by reprograming the RC substrate site. Therefore, the mitochondrial RC performs as an in vivo molecular oxygen sensor. Substantial differences between responses of rats with high and low resistance to hypoxia were determined.

## 1. Introduction

Hypoxia is an extremely widespread phenomenon, which causes various pathologies, including myocardium infarction, stroke, ischemia, hemorrhage, etc. In all cases, either oxygen delivery to the cell decreases to a level that is not sufficient to maintaining the cell function and metabolism or oxygen consumption and utilization by specialized systems become disordered.

According to current understanding, mitochondria play a signaling role in modulation of oxygen consumption by regulating the rate of extracellular oxygen delivery as well as by regulating oxygen homeostasis that is essential for normal vital activity [1,2,3,4,5,6,7,8,9]. In hypoxia, mitochondria contribute to formation of both immediate and long-term adaptive responses and, thereby, provide an integrated response to oxygen deficiency [5,10,11,12,13]. Clearly, the issue of oxygen homeostasis during adaptation to hypoxia is relevant and important.

At the present time, it has been convincingly proven that during adaptation to hypoxia, the cellular energy demand is met not only by modification of oxidative phosphorylation (OXPHOS), but also by kinetic regulation and changes in properties and contents of mitochondrial enzymes [12,13,14,15]. In this process, the respiratory chain (RC) substrate site (mitochondrial complexes, MC I-II) plays a special role [5,11,16]. However, systematic studies of the involvement of mitochondrial enzymes in responses to different regimens of acute or long-term hypoxia and the role of these enzymes in the formation of adaptive responses that influence hypoxia resistance have not been reported. This was the objective of the present study.

It should be noted that at the present time, molecular mechanisms of hypoxia are generally studied in the conditions of very severe O_2_ shortage that is not typical even for pre-anoxic or anoxic conditions (1–1.5% O_2_). However, it is well known that oxygen distribution in the cell is heterogeneous and is determined by the intracellular oxygen gradient, which is formed in the area of mitochondria [17,18,19,20,21,22] and is associated with mitochondrial clustering in the cell [23]. A combination of only two variable factors, clustering and respiration rate, may lead to an increase in the cell O_2_ gradient by two orders of magnitude. In addition, the functional and metabolic activity of the cell is crucial for the regulatory relationship between the oxygen delivery to the cell and the formation of the oxygen concentration gradient [13,24]. Thus, in vivo we deal with a broad range of various intracellular oxygen concentrations. However, the state and functioning of the mitochondrial apparatus and its operation under these conditions is poorly understood.

The solution to this problem is not only of scientific, but also of great practical importance, since it can be used in medical practice for developing molecular methods to assess the degree of hypoxic damage and strategies for anti-hypoxic defense of the body.

Therefore, this study focused on the dependence of mitochondrial enzyme response and mitochondrial ultrastructure on the severity and duration of various regimens of in vivo exposures to hypoxia. Specifically, we examined the effects of hypoxia on the cerebral cortex (CC), the most hypoxia-sensitive tissue, as well as, on the resistance of individual animals to oxygen deficit.

The results of this study provide evidence that the response of studied parameters to various O_2_ concentrations most efficiently reflects the RC performance under these conditions and represents a predictive criterion for the state of oxygen homeostasis.

## 2. Results

### 2.1. The Rate of RC Enzymes and Morphological Features of Mitochondria under Normoxic Conditions in the CC of Rats with High (HR) or Low Resistance (LR) to Hypoxia

In normoxic conditions (21% O_2_ in exhaled air), concentrations of MC I-V subunits differed in rats with different tolerance to hypoxia (Table 1), which was consistent with our previous data [25,26]. Contents of all enzymes were significantly higher in CC of HR than LR rats. The greatest difference was observed for the NDUFV2 subunit of MC I (Table 1). Content of other CC enzymes, including SDHA (MC II), were also higher in HR rats although to a lesser extent (Table 1).

Baseline quantitative differences in the content of enzymes in the RC substrate site reflect their greater or lesser contribution to tissue energy metabolism. For instance, the NDUFV2 subunit is a terminal iron-containing cluster of the catalytic module of NADH dehydrogenase, the major pathway for oxidation of NAD-dependent substrates of CC mitochondrial RC. In HR rats, the content of NDUFV2 subunit was 1.5 times greater than in LR. This may reflect a greater significance of NAD-dependent oxidation in the HR phenotype. The same was true for the content of SDHA subunit, an iron-sulfur cluster component of SDH (MC II). This enzyme oxidates succinate and becomes activated under conditions of increased functional loads, such as hypoxia and stress [5,11,27,28].

According to morphometry data for normoxia, the shape of mitochondria from both LR and HR rats varied from round to elongated (Figure 1). The total number of mitochondria per unit area (10 µm^2^), which is regulated by the balance of biogenesis and degradation, did not significantly differ in cortical neurons of two animal phenotypes (Table 2). The proportion of small mitochondria (0.14–0.25 µm) was also approximately similar in LR and HR rat, whereas the number of elongated (0.5–3 µm) mitochondria was 10% greater in HR rats compared to LR rats. The number of hyper-elongated mitochondria (4 µm and more) was also 3.7 times greater in HR rats (Table 2). At the same time, the area and perimeter of small and elongated mitochondria of LR rats were somewhat smaller (20–30%) than in HR (Table 3).

Small mitochondria that are characteristic of resting cells with low respiratory activity prevailed in the prefrontal CC of LR rats during normoxia. At the same time, the number of elongated and particularly spaghetti-like mitochondria was significantly greater in HR than in LR rats (Table 2). Elongated mitochondria are typical for conditions associated with increased ATP production, higher values of membrane potential and decreased production of reactive oxygen species (ROS) [29,30,31,32,33,34,35,36,37]. These mitochondria have more numerous cristae, increased ATP synthase dimerization and activity, higher effectiveness of OXPHOS and ability to distribute energy over longer distances and to control mtDNA inheritance. In addition, these mitochondria are protected from autophagic degradation. [30,33,34,38,39,40,41,42,43,44].

The soma, dendrites and synaptic terminals of cortical mitochondria from LR rats were characterized with electron-light matrix and low-dense packing of cristae (Figure 1A). In contrast, mitochondria of HR animals had condensed matrix and cristae (Figure 1B). The CC of both rat phenotypes contained some mitochondria with cristae oriented along the mitochondrial long axis. Percentage of these mitochondria was 21% in HR and 15% in LR rats.

The shape, number and size of cristae are also markers of mitochondrial functional activity, since these features reflect the distribution area of respiratory enzyme complexes and F1F0-ATP synthase. These cristae are located in the mitochondrial membrane and represent major functional units related with energy conversion [45,46,47]. An increase in crista number indicates increased functional demand of the cell. Moreover, the crista shape determines the assembly and stability of RC supercomplexes and, hence, the rats’ efficiency of mitochondrial respiration [48]. Thus, in normoxia, the CC of LR and HR displayed distinct phenotypic differences in the content of catalytic subunits of mitochondrial enzymes and ultrastructure.

### 2.2. Effects of the Mild Hypoxic Exposure (Hypobaric Hypoxia, HBH, 14% O_2_; O_2_ Concentration in Air Decreased by 33%)

In CC of both rat phenotypes, the first hypoxic exposure as a part of long-term regimen, did not induce any significant immediate changes in concentrations of the NDUFV2 (MC I) and SDHA (MC II) enzyme subunits in the RC substrate site. The SDH activity remained unchanged in this process (Figure 2A,B).

Despite the absence of considerable changes in the response of RC substrate site enzymes, the content of Cyt b (MC III) and COX1 (MC IV) subunits was significantly increased in both animal phenotypes already by 15 min after the onset of mild hypoxia. This reflects potentiation of the electron transport function of the RC cytochrome site (Figure 2B).

Immediate changes also occurred in mitochondrial morphometric parameters. A single, one-hour, mild hypoxic exposure significantly increased, by 2.7 times, the number of small mitochondria with higher matrix density and more dense packing of cristae in LR rats. This brought them structurally more similar to the mitochondria of control HR animals. The number of elongated mitochondria remained practically unchanged, whereas the number of hyper-elongated (spaghetti-like) mitochondria slightly increased (Table 4). All of these changes reflect enhanced mitochondrial metabolic activity and potentiated OXPHOS [3,31,32,44,48,49].

In contrast, in HR rats, the number of small mitochondria, which was significantly greater than in control LR, increased only 1.3 times after this hypoxic exposure. Furthermore, there were no changes in the number and the density of cristae packing. However, during this period, the amount of elongated and hyper-elongated mitochondria was significantly decreased. This indicated intensified mitochondrial fragmentation (Table 4).

Therefore, the regimen of “mild” hypoxic exposure induced, primarily in LR animals, immediate mitochondrial fragmentation with increased metabolic activity. This is a typical mitochondrial response to any external stimulus.

The long-term hypoxic exposure of LR animals was associated with a brief increase in the amount of SDHA subunit (25% after session 3) and a decrease in the SDH activity, which recovered by session 20. On the contrary, the amount of the NDUFV2 subunit increased slightly after the 8th hypoxia session by 10–20% (Figure 2A, Table 1). All of these processes reflect a gradual decrease in the contribution of succinate oxidase oxidation in LR rats during this period while the contribution of NADH oxidase oxidation was maintained and/or increased.

In contrast to LR rats, there were no significant changes in the content of NDUFV2 subunit in HR rats during the entire period of long-term hypoxia. Moreover, in HR rats, the increase in SDHA subunit expression during the same period was not associated with changes in the SDH activity, which remained at the normal level practically throughout all hypoxic exposures (Figure 2B).

At the same time, the high level of Cyt b persisted both during the early post-hypoxic period and for 20 sessions. However, the level of COX1 normalized in LR rats after session 8 and in HR rats after session 3.

As for the significant time-related changes in the content of the ATP5A subunit, an index of ATCase activity, these changes were not observed in HR rats during the hypoxic exposures, except for small fluctuations. Only after session 20, the amount of ATP5A subunit decrease (by 20% from baseline, Figure 2B). In contrast, in LR rats a small increase (up to 20%) in the amount of this enzyme was observed. This increase may reflect enhanced ATP-synthesizing activity during this period followed by subsequent normalization (Figure 2A). However, at the end of the long-term hypoxic exposures, the amount of COX1 and ATP5A subunits decreased from baseline by 15–20% in both cases (Figure 2A,B, Table 1). This may reflect transition to a new, more economical level of energy regulation. This is a sign of adaptation to the hypoxic regimen.

This conclusion is also supported by significant transformation of the mitochondrial ultrastructure in LR and HR animals by the end of the hypoxic exposures. In LR rats by the end of long-term hypoxic exposure, the number of small mitochondria even though it sharply decreased compared to the first exposure, had doubled compared to the control value. The number of hyper-elongated mitochondria remained small. However, the number of elongated mitochondria with condensed matrix was significantly increased (Table 4, Figure 3A). This indicated enhanced mitochondrial fusion activity, an index of optimized mitochondrial function.

In contrast, in HR rats, by the end of long-term hypoxic exposures, the number of small and elongated mitochondria, returned to the baseline level, whereas the number of hyper-elongated mitochondria remained 35% reduced (Table 4, Figure 3B). The packing density and number of cristae did not differ from the control values. There were no signs of swelling, but the volume of matrix was increased compared to the volume of cristae. Thus, the ultrastructural changes observed during this process were not pronounced.

### 2.3. Effect of a Course of “Moderate” Hypobaric Hypoxia (HBH, 10% O_2_; O_2_ Concentration in Inspired Air Decreased by 50%)

The response of mitochondrial enzymes to this regimen of hypoxic exposures was considerably different from that observed with the mild one. The first hypoxic exposure induced rapid (within 15 min), simultaneous and reciprocal changes in contents of enzyme subunits in the RC substrate site of both HR and LR rats: (1) There was significant decrease in the content of NDUFV2 subunit by the end of hypoxic exposure (30–35% decrease in LR and 15–20% decrease in HR), which reflected inhibition of MC I electron-transport function. (2) In addition, a simultaneous, immediate increase (15–20%) in the content of SDHA subunit was accompanied by an increase in SDH activity, which indicated activation of the MC II. (Figure 4A). Thus, in this instance, MC II became the major electron supplier for the RC.

The simultaneous, immediate expression of Cyt b and COX1 subunits sharply increased by 40–70% and 20–40%, respectively, compared to the control and the mild hypoxic regimen (Figure 4B). This suggests that the FAD-dependent electron flow formed by succinate oxidation up-regulates the electron-transport activity of the RC cytochrome site, as compared to the oxidation of NAD-dependent substrates. The ability of the succinate oxidase pathway to monopolize the RC at the expense of a higher velocity of substrate oxidation and, furthermore, to induce the reverse electron transport, was shown for the first time by Chance [50]. Nevertheless, during that period, the level of ATP5A was either unchanged, as in LR rats or even decreased as in HR rats. This could have been due to a reduced protonic gradient during succinate oxidation. This decrease in the protonic gradient may reduce proton translocation from the intermembrane space into the matrix via ATP synthase (MC V) that catalyzes ADP conversion to ATP.

Immediate morphological alterations of mitochondria in response to the first hypoxic exposure were similar to that observed for the mild regimen but were more pronounced quantitatively. The mitochondrial fragmentation that characterizes the mitochondrial transition to the active metabolic state was more pronounced in LR rat (Table 4). Furthermore, in LR rats, the population of small, electron-dense mitochondria was increased by almost 4-fold. However, the amount of such mitochondria in HR rats, which was quite large in control rats, remained unchanged during this moderate hypoxic exposure (Table 4). In both animal phenotypes, the amount of elongated mitochondria did not significantly differ from control values or those of mild hypoxia.

In both animal phenotypes, under the conditions of moderate, long-term hypoxia, immediate changes in enzyme levels persisted during the first few sessions. The content of SDHA subunit started to decrease after session 3, and it was reduced by 50–60% from baseline by the end of hypoxic exposures. This was associated with a decrease in SDH activity. In LR rats, the SDH activity was decreased after the first session, but it stabilized after session 8, and then it remained at the baseline level until the end of hypoxic exposures. In HR rats, the SDH activity was decreased by 15–20% compared to control at 24 h after the first hypoxic exposure and remained at this level until the end of the exposures (Figure 4A, Table 1). The content of NDUFV2 subunit remained reduced until session 8. However, subsequently it not only completely recovered by session 20 but even exceeded the baseline level by 10–20% (Table 1, Figure 4A). Thus, the activation of MC II during long-term, moderate hypoxia was relatively brief. Eventually, this process created conditions for recovery of the MC I function.

As for the cytochrome site, of both animal phenotypes, when the content of NDUFV2 was decreased, concentrations of Cyt b and COX1 subunits remained increased through session 8 to a greater extent than in mild hypoxia. The content of ATP5A subunit also increased during that period, especially in LR rats. Therefore, long-term hypoxic exposure to 10% O_2_, despite the change in the ratio of metabolic flows in the RC substrate site, was associated with a sharp increase in the content of cytochrome site enzymes during sessions 1–8. This demonstrated enhancement of the electron-transport and energy-producing function during this transitional, adaptive period. These facts confirm the ability of succinate oxidase oxidation to maintain high activity of the electron transport function of the RC cytochrome site during long-term hypoxia.

After completion of the long-term exposure to moderate hypoxia, the contents of COX1 and ATP5 decreased by approximately 10% from baseline (Figure 4B, Table 1). This may indicate that adaptation of these enzymes to moderate hypoxia had been completed.

The mitochondrial ultrastructural pattern also changed by the end of long-term hypoxia exposure. In LR rats, the number of small mitochondria decreased compared to the immediate response to the first hypoxic exposure, although it doubled compared to the baseline values. The number of elongated mitochondria without signs of swelling, with condensed matrix and tightly packed cristae increased, which optimized mitochondrial function in these conditions. Spaghetti-like and ramified mitochondria appeared as shown in Figure 5A. The number of hyper-elongated (spaghetti-like) mitochondria also slightly increased. This reflects enhanced mitochondrial metabolic activity and potentiated OXPHOS.

### 2.4. Effects a Course of the “Severe” Hypobaric Hypoxia (HBH, 8% O_2_; O_2_ Concentration in Air Decreased by 62%)

The response of mitochondrial enzymes in the RC substrate site to severe hypoxia was essentially different from that observed in the previous case. The first hypoxic exposure caused immediate, simultaneous expression of NDUFV2 and SDHA subunits at 15–30 min without SDH activation, which was more pronounced in LR rats. Thus, the response to severe hypoxia was mobilization of both metabolic flows of the RC substrate site, i.e., NAD-dependent (MC I) and succinate-oxidase (MC II) (Figure 6A). However, the level of both subunits began to decrease already during the 1st exposure. These changes persisted for 24 h.

Despite the immediate, simultaneous activation of MC I and MC II at the cytochrome site, the immediate changes were considerably less pronounced than in the two previous regimes. There was a brief and small activation of the Cyt b and ATP5A subunits, but there were no changes in the amount of COX1 subunits.

Changes in CC mitochondrial ultrastructure in response to the first severe hypoxic exposure also differed from effects of the previous two hypoxic regimens. In LR rats, the number of small mitochondria was increased compared to the control, but this number was almost a half of that observed in moderate hypoxia (Table 4). In HR rats, the number of small mitochondria decreased by more than 50% compared to the control value. Thus, in this case, the mitochondrial fragmentation typical for the immediate response to an external stimulus sharply decreased. Moreover, spaghetti-like, hyper-elongated mitochondria with tightly packed cristae, which could reach a length of 3–5 µm and which were not typical for this period, appeared in CC of both LR and HR rats. In LR rats, these mitochondria had a more electron-dense matrix than did those mitochondria of HR rats. The latter process may be associated with increased formation of the mitochondrial reticulum. This would provide more economical cell metabolism under severe hypoxia and improved regulation of the membrane potential [33]. These mitochondria of LR rats had a more electron-dense matrix than did the mitochondria of HR rats. Thus, the changes in the structure of mitochondria during severe hypoxia were more pronounced than during mild and moderate hypoxia.

As the course of hypoxic exposures continued, the contents of NDUFV2 and SDHA subunits reached minimum values at soon as by session 3 in both rat phenotypes. However, immediately afterward, the content of NDUFV2 subunits began to recover, and after the 8th session, this content exceeded the initial values by 15–20% for LR rats and by 5–10% for HR rats.

In contrast, the content of SDHA subunits remained low until the end of the long-term hypoxic exposure. In LR rats during sessions 3–8, this content was 20–25% lower than baseline values, and it was not more than 10% lower by the end of long-term hypoxia. The dynamic changes in SDH activity were similar. In HR rats, the decrease in SDHA content was greater, and it reached 40% during sessions 8–12 with no changes in SDH activity, which had stabilized at a normal level by the end of the first session (Figure 6A, Table 1). This indicates a sharp decrease in MC II activity by the end of the long-term exposure. Thus, in severe hypoxia, the activation of MC II was very brief, i.e., 1–3 days after the first hypoxic exposure and the recovery of MC I function had started already by completion of session 3. This was considerably earlier than observed for moderate hypoxia.

Despite the simultaneous activation of MC I and MC II during this hypoxic regimen, the response of cytochrome site enzymes was considerably more moderate compared to the previous regimen (10% O_2_). Thus, the content of COX1 subunits in LR and HR rats remained within a normal range, after the first hypoxic exposure and during the subsequent 12 sessions. However, after session 20, the content of COX1 subunits decreased by 20% (Figure 6B). The increase in ATP5A subunits was observed in both rat phenotypes only during the first hypoxic session and on the following day. After session 3, the content of this enzyme had decreased to the baseline value; after session 20, it had decreased by 40% from control values (Figure 6B). This may reflect a sharp decline in the mitochondrial ATP-producing function. The content of Cyt b, such as in previous hypoxic regimens, was significantly increased already at 15–30 min after onset the hypoxia, although this increase was less than during mild and moderate hypoxic exposures, and it stabilized at this level until the end of long-term hypoxia. Understanding these dynamics of Cyt b will require further study.

By the end of long-term hypoxia, the numbers of small and spaghetti-like mitochondria from the CC of LR and HR rats were decreased practically to control values. However, in LR rats, the number of spaghetti-like mitochondria, whose length reached 3–5 µm, sharply increased (Table 4). In HR rats, this severe hypoxic exposure induced formation of two more mitochondria shapes: (1) ring-shaped mitochondria (donut-shaped) and (2) blob-shaped mitochondria without signs of swelling, with condensed matrix and densely packed cristae [44] (Figure 7A,B). The emergence of these forms reflects a peculiar adaptation of mitochondrial morphology to the sharp oxygen deficit that occurred during long-term hypoxic exposures.

### 2.5. Effects of Different Long-Term Hypoxia Regiments on Dynamics of Rats Resistance

Baseline resistance of LR and HR rats to acute hypoxia (Tr) was assessed at the systemic level by the time of “survival” in the altitude chamber during exposure to a subcritical altitude (see Methods).

During long-term exposure to mild hypoxia (14% O_2_), Tr values did not significantly change in either LH or HR rats after session 8 or 20. Thus, this long-term hypoxic regimen resulted in a slight increase in resistance only in LR rats (Figure 8). However, during more severe, long-term hypoxia (10% and 8% O_2_), the resistance of LH increased 2.5 times as soon as after session 8 (Figure 8). In contrast, in HR rats, the long-term moderate hypoxia (10% O_2_) did not significantly influence the Tr values; however, Tr doubled in severe hypoxia (8% O_2_) already after session 8 (Figure 8). Therefore, the long-term resistance developed at the systemic level in LH rats to either moderate or severe hypoxia, whereas in HR rats, this resistance developed only in severe hypoxia.

## 3. Discussion

Comprehensive studies of mitochondrial function and ultrastructure were conducted in this investigation. These studies included measuring the content of mitochondrial enzymes and structural and functional features of mitochondria in the CC of two rat phenotypes with different tolerance to acute repeated hypoxia. The measurements were performed under the conditions of normoxia or oxygen deficit. The study revealed considerable differences in the performance of the mitochondrial energy-producing apparatus, both under normoxic conditions and during various regimens of hypoxia.

Earlier we described basic, systemic differences in functional and metabolic patterns between LR and HR rats during normoxia [11,25,26]. In the present study we demonstrated differences between HR and LR rats in contents and kinetic properties of RC enzymes of brain tissue, a target for hypoxia and in mitochondrial ultrastructure. These data also support the notion of significant differences in the performance of the energy apparatus in response to hypoxia. Thus, in normoxia, CC mitochondria of HR and LR rats are in different functional and metabolic states, as evidenced by their bioenergetic and ultrastructural indexes. This suggests that the observed phenomenon is genetically predetermined, and that the energy metabolism is a leading factor in development of individual resistance to hypoxia. Genetically determined differences in the catalytic activity of mitochondrial respiratory complexes and supercomplexes have been reported for various murine lines [51]. This implies various possibilities of their participation in the regulation of RC electron transport function and their influence on the effectiveness of RC performance and ATP synthesis.

According to current concepts, a hypoxic exposure (single, repeated or chronic) induces formation of immediate and long-term adaptation. Immediate adaptation forms after the onset of exposure to environmental stress and is based on available, preformed physiological mechanisms and programs. Long-term adaptation develops gradually, as a result of lengthy or repeated exposure to stress. This adaptation is not based on presently available physiological mechanisms but rather on newly formed regulatory programs [52]. In result, oxygen homeostasis and energy metabolism transfer to a new regulatory level that results in a more economical spending of energy and increased metabolic power following the conditions of activation. In sum, both of these processes aim to optimize ATP synthesis during hypoxic conditions. However, it should be taken into account that too mild hypoxia may not activate mechanisms of immediate or long-term adaptation. These mild conditions would induce only transient changes within a physiological range of responses to decreased biological oxidation. In contrast, excessively intense hypoxia may cause maladaptation, disorders of functions and metabolism and damage to structures.

Our results show that the substrate site (MC I-MC II) and cytochrome site (MC III-MC IV) of the CC RC can mobilize crucially different types of response either to immediate changes in ambient O_2_ or to the conditions of long-term exposures to various hypoxic regimens.

The oxygen sensitivity threshold in rats as assessed by physiological indexes at the systemic level is considered to be approximately 12% O_2_ relative to normal atmospheric 21%. Elevation to an altitude of 13,000–14,000 m in the HBH conditions is critical for rats and results in 100% lethality. Taking this into account, we used three different hypoxic regimes corresponding to mild, moderate and severe hypoxia as described above. Depending on the severity of hypoxic exposure and animal phenotype, we identified three major types of mitochondrial enzyme responses at both RC substrate and cytochrome sites.

The first type of response was typical for the “mild” hypoxia (14% O_2_ in inhaled air, 34% lower than atmospheric). In response to one-hour hypoxic exposures in this regimen, there were no substantial immediate changes in the content of NDUFV2 and SDHA subunits in LR and HR. However, already after 15 min, there was a significant increase in amount of Cyt b (MCIII) and COX1 (MCIV) subunits, which reflects potentiation of the electron transport function of the RC cytochrome site. The obtained data indicate that CC mitochondrial RC differentiates even a mild hypoxic exposure by urgent transition to the active state without considerably changing the pathway of energy substrate oxidation, i.e., at the expense of physiological reserve (predominant oxidation of NAD-dependent substrates characteristic of normoxic CC remains).

Immediate changes also occurred in mitochondrial morphometric parameters. A single, one-hour, mild hypoxic exposure significantly increased the number of small mitochondria with higher matrix density and more dense packing of cristae in LR rats, which structurally brought them closer to mitochondria of control HR animals. The number of hyper-elongated (spaghetti-like) mitochondria also slightly increased. This reflects enhanced mitochondrial metabolic activity and potentiated OXPHOS [3,31,33,44,48,49].

In HR rats, the increase in number of small mitochondria was minor. However, the number of elongated and spaghetti-like mitochondria significantly decreased in this period also indicating intensified mitochondrial fragmentation.

The long-term hypoxic training of LR animals (14% O_2_) was accompanied by only small changes in levels of RC substrate site enzymes, which reflected a gradual decrease in the activity of MC II and an increase in the contribution of NAD-dependent oxidation. However, the increased level of RC cytochrome site enzymes remained.

A high level of Cyt b persisted both through the early post-hypoxic period and through 20 sessions of hypoxia. The increased level of COX1 returned to normal in LR rats after session 8, and in HR rats after session 3. The ATCase activity in HR remained at the baseline level, and in LR it even periodically increased. Thus, in the long-term hypoxic exposure, there were no disorders in the electron transport function of the RC cytochrome site.

Nevertheless, at the end of long-term hypoxic exposures, the content of COX1 and ATP5A subunits decreased, which may reflect transition to a new, more economical level of energy regulation as a sign of adaptation to this hypoxic regimen. This conclusion is also supported by the significant transformation of mitochondrial ultrastructure in LR animals that was evident by the end of long-term hypoxic exposures.

The increased number of elongated mitochondria with condensed matrix makes LR mitochondria similar to those of HR rats. This reflects enhanced mitochondrial fusion activity, an index of optimized mitochondrial function. This includes stimulation of bioenergetic processes, ATPase activation and potentiation of the ATP-producing function in these mitochondria [3,31,41,45,49]. Mitochondrial fusion is a process that facilitates the communication between mitochondria and the host cells [53]. In long-term hypoxia, this mitochondrial fusion reflects enhancement of regulatory, adaptive mechanisms that are responsible for the quality of the total mitochondrial population.

In contrast, in HR rats, such ultrastructural changes were absent at the end of long-term hypoxia. Moreover, the number of elongated and spaghetti-like mitochondria decreased significantly during this period, which indicates an increase in mitochondrial fragmentation. Therefore, long-term adaptation to mild hypoxia is associated with formation of a mitochondrial population with reduced content of enzymes in the RC terminal site but working in a more economical mode. However, a new ultrastructural pattern was formed only in the CC of LR rats.

It should be noted that in long-term mild hypoxia, Tr, an index of in vivo resistance to oxygen deficit at the systemic level, remained practically unchanged in LR and HR rats after both sessions 8 and 20 (Figure 8). However, dynamic changes in mitochondrial enzymes showed that mitochondria had a higher oxygen sensitivity threshold and were able to differentiate considerably smaller changes in oxygen concentration than Tr, i.e., performed as sensitive, intracellular oxygen sensors.

Earlier we showed that throughout the entire period of mild hypoxia, the glutathione system of both rat types retained its regulatory capability for maintaining the reduction capacity of the cell characteristic of normal, physiological state, without increased free-radical activity [54]. These data exclude a possible influence of ROS on the function of RC in this hypoxic regimen. The decreased ROS formation and the increased GSH reduction during mild hypoxic regimens have been reported also by others [55,56,57,58,59].

The second type of response was typical for the moderate HBH (10% O_2_; 50% decrease in O_2_ concentration compared to atmospheric air). The main findings in rats exposed to long-term HBH were: (1) Change in oxidation substrates in CC mitochondria in response to the first and subsequent hypoxic sessions. Specifically, there was depression of the MC I function and activation of MC II; (2) Restoration of MC I function by the end of long-term hypoxic exposure that was associated with a decrease in MC II activity, i.e., a decline in the content of SDHA subunit and SDH activity.

The beneficial effect of succinate on the recovery of respiration and the OXPHOS system during hypoxic conditions was discovered as early as in 1948 [60]. Under the conditions of hypoxia, this pathway has thermodynamic advantages over the oxidation of NAD-dependent substrates. Despite the fact that only two phosphorylation points remain in this process, high velocity of this reaction provides a high energetic efficiency of the entire process. Multiple reports have confirmed this effect and a special role of succinate in physiological processes [27,28,61,62,63]. However, we were the first to describe the hypoxic inhibition of MC I function associated with compensatory activation of succinate oxidase oxidation [11,16]. We also have shown that in this process, the kinetic parameters, Vmx and Km, of cerebral NADH-cytochrome c reductase and COX changed. The enzyme activities increased, whereas their substrate affinity decreased [12,13,14,15,64]. As a result, these enzymes of hypoxia-adapted animals acquired a capability for functioning over a broader range of substrate concentrations and also more efficiently. Indeed, after long-term adaptation to this hypoxic regimen, the oxidation velocity of NAD-dependent substrate decreased and in LR rats to a greater extent that in HR rats. However, the effectiveness of phosphorylation as estimated by the ATC/O ratio, as well as ATP synthesis, was in this process higher that before adaptation [11,12,14,15,64,65,66]. The free-radical activity in CC of both animal types also remained within a norm al range [54]. Later, it was established that, despite the hypoxic inhibition of MC I activity, the MC III and MC IV activities and the normal values of ΔΨm and ATP are maintained due to the electron flow from MC II. A succinate shortage leads to reversible loss of the membrane potential, which can be easily restored by succinate supplementation [67].

The activation of MC II in this hypoxic regimen, which correlates with the depressed MC I function, is an immediate regulatory and compensatory molecular mechanism of adaptation, which occurs in vivo. This mechanism helps to prevent or to alleviate disorders of ATP synthesis typical of hypoxia, to improve parameters of the adenylate pool and vital functions of the body, to stabilize and improve pH and to abolish hypoxic acidosis [5,11,14,16,61,68,69]. Moreover, exactly during the period of MC II activation, MC I catalytic properties are modulated, which makes possible the recovery of its function and reflects adaptation of mitochondrial enzymes to this hypoxic regimen.

This conclusion is supported also by the direction of structural and morphological changes in mitochondria by the end of this long-term hypoxia. These changes were much more pronounced in LR rats and they included enhanced mitochondrial fusion, condensed matrix and cristae, absence of swelling and emergence of elongated mitochondria. The latter process may be related with increased formation of the mitochondrial reticulum, which would provide more economical cell metabolism in that period and improved regulation of membrane potential [33].

The emergence of ramified mitochondria, which are one of forms of mitochondrial aggregation, is considered to result from their reduced motility. These mitochondria do not lose their membrane potential; they are characterized by a low level of free-radical activity, and by an increased number of nano-tunnels, which connect them and facilitate intermitochondrial metabolism [70]. These mitochondria may be a result of mtDNA mutations and are able to resist autophagosomal degradation [71]. Such mitochondrial population has increased resistance to oxygen deficit and retains its ability to function actively in these conditions. All these features reflect adaptation of mitochondrial morphology in CC of LR rats to oxygen deficit during hypoxic exposures.

In contrast, in CC of HR rats exposed to long-term hypoxia, the number of small mitochondria was still only slightly higher than in control whereas the number of spaghetti-like mitochondria was decreased (Figure 5B). The number of elongated mitochondria remained within the normal range. However, the volume of their matrix was increased more than was the crista volume and this difference was greater than in mild hypoxia. This suggests that biosynthetic processes were potentiated [49]. Thus, also in this instance, adaptive morphological changes in mitochondria of HR rats were very weakly pronounced. Light mitochondria almost completely disappeared in CC of both rat types.

Thus, in this hypoxic exposure (10% O_2_), three periods with different time-related changes in CC mitochondrial enzymes of both animal phenotypes were distinguished: (1) formation of immediate mechanisms of adaptation to hypoxia (immediate response to the first hypoxic exposure). (2) Formation of mechanisms of long-term adaptation to this exposure (3–12 hypoxic exposures). (3) Completion of the formation of long-term adaptation by the 20th exposure. This process results in emergence of a mitochondrial population with (1) new kinetic properties, which allows maintaining high effectiveness of OXPHOS; and (2) ultrastructural changes in mitochondria, which reflect various degrees of mitochondrial energization.

Evaluating of a possibility of systemic adaptation to this 10% O_2_ hypoxic regimen showed that Tr values increased 2–2.5 times by of long-term hypoxic exposure in LR rats. However, in HR rats, the resistance to acute hypoxia either remained unchanged or changed to a considerably less extent, which was consistent with our earlier results [5,11,12,13].

Thus, the dynamics of mitochondrial enzymes and the ultrastructure of mitochondria are highly sensitive molecular indicators of the adaptation of energy metabolism to oxygen deficiency.

The third type of enzyme response in the RC substrate site was characteristic of long-term severe hypoxia (8% O_2_; decrease in O_2_ concentration by 60–63% compared to atmospheric air). In this case, three periods were also identified, and these periods were also characterized by specific dynamic and in mitochondrial ultrastructure. However, these changes were significantly different from the moderate hypoxia. Thus, the first severe hypoxic exposure resulted in a simultaneous activation of MC I and MC II, which was either very brief and was observed in LR only during the first one-hour hypoxic exposure and in HR during the first day. However, already by the end of session 1, after a brief normalization of the contents of NDUFV2 and SDHA subunits during sessions 1–3, we observed reciprocal changes in the contents of SDHA and NDUFV2 subunits, i.e., decreased SDHA and increased NDUFV2 subunits. These changes were similar to those observed during the moderate hypoxic regimen and may reflect the completion of the succinate-dependent recovery of MC I activity, which in this case proceeds very quickly, but is much less pronounced than during moderate hypoxia.

The mechanism responsible for recovery of the MC I activity in acute ischemic, pre-anoxic and anoxic conditions, are presently known. All of these mechanisms provide reduction of this complex activity and include reversion of the tricarboxylic acid cycle with formation of succinate and the coupled succinate-dependent activation of aminotransferase reactions, substrate phosphorylation of α-ketoglutarate, α-glycerophosphate and the activation of the purine nucleotide cycle [10,27,28,61,62,68,72,73,74,75]. All these reactions facilitate a decrease in reduction of pyridine nucleotides and, thereby, the recovery of MC I function.

Despite this, the response of cytochrome site enzymes was considerably more moderate compared to the previous regimen. So, after sessions 12 both the COX function and ATPase activity the declined sharply. This indicates a decrease in the ATP-synthesizing function and the MC IV activity. Noteworthy, in this regimen, free-radical processes were activated already during the first hypoxic exposure and continued growing up to the end of the course of training, which might affect the RC function. [54].

Changes in mitochondrial ultrastructure in response to the first hour of severe hypoxia also differed from the effects of the previous two hypoxic regimens. The fragmentation of mitochondria characteristic of the immediate response to an external stimulus was abruptly decreased in this case. Moreover, in the CC of both LR and HR rats, spaghetti-like mitochondria with tightly packed cristae emerged, which could reach 3–5 µm. Such mitochondria were absent during this period under other hypoxic regimens. In LR animals, these mitochondria had more electron-dense matrix than mitochondria of HR rats. Such mitochondria are considered as a sign of reduced mitochondrial fission, and as such, they are a means of protection against moderate oxidative stress [37,76,77].

At the end of the hypoxic exposures, the numbers of small and elongated mitochondria in LR and HR had decreased practically to control values. At the same time, in LR rats, the number of spaghetti-like mitochondria was increased. Simultaneously, mitochondrial spheroids, i.e., “donuts”, appeared, which represented further transformation of spaghetti-like mitochondria curling up into a ring. This shape is a reversible, intermediate state that forms under decreased respiration and a dramatic ATP deficit. This shape forms also under conditions of increased ROS production, dysregulation of calcium metabolism and increased K+ conductivity of the inner mitochondrial membrane [76,77,78]. When conditions are optimized, the “donuts” break up into several linear structures and they are able to restore ΔΨm. However, if metabolic stress continues, they may undergo autophagy. Thus, the mitochondrial transformation into “donuts” may represent a component of a protective mechanism to help preserve mitochondria under conditions of progressive energy shortage and free-radical activity [19,44,76,77,78,79].

Another shape characteristic for this period was huge, bag-like or drop-like mitochondria with condensed matrix and tightly packed cristae such as those described by Ahmad et al. [21]. According to these authors, these mitochondria may also be characterized by low ΔΨm values, and increased resistance to autophagosomic degradation, and disorders of calcium metabolism. Both shapes of mitochondria (“drops”/“donuts”) were suggested to be predictors of oxidative stress; the “donut” shape may be its early marker while the “drop” shape may be a marker of irreversible toxicity induced by increased ROS production [76].

On the whole, the pattern of mitochondrial dynamics at the end of this course of hypoxia indicates gradually progressing symptoms of RC dysfunction, although long-term adaptation did develop in both animal phenotypes at the systemic level. Therefore, also in this case, coupled changes in mitochondrial enzymes and mitochondrial dynamics are the most sensitive indexes of the RC state during development of mechanisms for adaption to oxygen deficit.

Based on analysis of these data we can conclude that three mechanisms are involved in the response to acute and repeated hypoxic exposures: (1) modulation of the content of mitochondrial enzymes; (2) modification of their kinetic properties; (3) mitochondrial dynamics. This reflects a possibility of very subtle and fast modulations of the state of the enzyme that depend on the O_2_ concentration and the duration of hypoxia. Undoubtedly, they influence formation of both immediate and long-term adaptive reactions. Thus, according to the dynamics of mitochondrial enzymes, the energy metabolism became more economical during long-term hypoxic exposures. This was evident already at 14% O_2_, reached a maximum at 10% O_2_ and declined at 8% O_2_.

Of special theoretical and practical interest are the direct proofs obtained in this study of a possibility of relatively brief (one or several hours or days) transition to succinate-oxidase oxidation in a definite range of reduced O_2_ concentrations. As mentioned above, this pathway plays a role of an immediate compensatory mechanism that provides preservation of the cytochrome site function and ATPase activity as well as facilitates changes in MC I kinetic properties. In our experiments, the maximum effect of such reprograming was obtained at 10% O_2_, whereas at 8% O_2_ this effect dramatically decreased.

Nevertheless, the molecular mechanism of RC reprograming and changes in metabolic fluxes (NADH-oxidase to succinate oxidase) is still undefined and requires further investigation. Such study is particularly relevant in the light of new data that MC III and MC IV may be dynamically organized into supercomplexes due to the FAD-dependent electron flux [80,81,82,83]. Moreover, there is evidence that MC II may play a regulatory role in the formation and maintenance of the respirasomes integrity [84,85,86,87,88,89]. It has also been reported that succinate oxidation is controlled by the adrenergic regulation, and the SDH activity is its marker [90].

Thus, CC mitochondria effectively and subtly differentiate gradual changes in the oxygen concentration in inspired air. In this process, the responses that mitochondrial enzymes develop are specific for each oxygen regimen. These responses reflect essentially different mechanisms for reprogramming the function of RC in ways that correlates with changes in mitochondrial ultrastructural pattern. According to current concepts, changes in mitochondrial ultrastructure are one of the ways for induction of intracellular signaling [31].

## 4. Conclusions


In in vivo conditions, gradual changes in reduced oxygen content result in CC rats a differentiated response of MC I-V enzyme subunits and in mitochondria ultrastructural changes. Together these parameters play a role of highly sensitive molecular markers reflecting the state of energy metabolism and the formation of mechanisms of immediate and long-term adaptation of the mitochondrial apparatus to oxygen deficit.The MC II activation in a certain range of reduced O_2_ concentrations is a compensatory mechanism, that ensures the preservation of the activity of RC cytochrome site enzymes, the formation of new kinetic properties of MC I and the restoration of its activity under these conditions, as well as the ultrastructural reorganization of mitochondria.The level of mitochondrial enzymes and mitochondrial dynamics in the CC LR and HR rats are different and affect the formation of urgent and long-term adaptation of animals to hypoxia.


## 5. Materials and Methods

### 5.1. Materials

All chemicals were purchased from Sigma-Aldrich, St. Louis, MO, USA. Primary polyclonal antibodies to NDUFV2 (sc-324161), SDHA (sc-27992), cyt b (sc-11436), COX1 (sc-23982), ATP5A and secondary antibodies (sc-2030, sc-2768) conjugated with horseradish peroxidase were purchased from Santa Cruz Biotechnology, Dallas, TX, USA.

### 5.2. Evaluation of Resistance of Animals to Acute Hypoxia

Experiments were performed on outbred rats with different baseline resistance to oxygen shortage. Tolerance of acute hypobaric hypoxia (HBH) was evaluated one month prior to the experiment [91,92]. The critical, life-incompatible altitude for rats was 13,000–14,000 m. We evaluated the ability of rats to survive in the altitude chamber at a subcritical altitude of 11,000 m by determining Tr (resistance index), to time of apnea onset.

After reaching Tr, the pressure in the chamber was normalized to the “sea level”, and the animals restored the normal posture and locomotor activity within 4–6 min. The Tr value was 1–2 min for control LR rats and more than 8 min for control HR rats. Typically, in a sample of 100 rats, 30–35% were LR to acute hypoxia (Ts < 2 min), 20–25% were HR (Ts > 6–8 min) and 40–50% were mid-resistant (Ts = 3–5 min).

### 5.3. Hypoxia Regimens

Experiments on forming immediate and long-term adaptation to hypoxia included various HBH regimens to simulate the phenomenon of high altitude, which is well known for its beneficial effect on the body. Rats begin “feeling” oxygen shortage, when the ambient O_2_ concentration decreases to 12%. Hence, this study used HBH models of different severity: (1) *mild* (subthreshold) hypoxic exposure (HBH 523 torr equivalent to FiO_2_ 14% O_2_ or elevation of 3000 m); (2) *moderate* hypoxic exposure (HBH 380 Torr, equivalent to FiO_2_ 10% O_2_ or elevation of 5000 m; and (3) *severe* hypoxic exposure (HBH 290 Torr, equivalent to FiO_2_ 8% oxygen or elevation of 7000 m).

Single hypoxic exposure included a one-hour daily exposure of animals in the altitude chamber. A long-term hypoxic exposure consisted of 20 sessions.

The control group was kept outside the hypobaric chamber in the same location. Effects of a single, one-hour hypoxic exposure or *long-term* hypoxic exposures (one hour daily for 20 days) to each of the three HBH regimens were studied. After the experiment, all rats were alive and resumed their normal activity without any sign of pathology.

The experimental procedures employed in this study agree with the principles and practices of the 1986 European Guide for the Care and Use of Laboratory Animals in accordance with the Ethical Guidelines from the European Community Council Directive (86/609/EEC) and published in the Order of the Ministry of Healthcare of the Russian Federation from 19 June 2003 no. 267.

### 5.4. Extraction of Proteins from CC Tissue

After decapitation of rats, the brain was rapidly excised and one part was immediately frozen in liquid nitrogen and stored at −80 °C. Prior to biochemical studies, the BC was homogenized in liquid nitrogen to powder. Two buffers were used for extraction of cytoplasmic proteins: (i) 20 mM Tris-HCL, pH 7.5; 1 mM EDTA; 1 mM dithiothreitol; 1 mM Na_3_VO_4_; 1 mM AEBSF; 60 μg/mL aprotinin; 10 μg/mL leupeptin; 1 μg/mL pepstatin A and (ii) 50 mM HEPES, pH 7.6; 150 mM NaCL; 2 mM EGTA; 1% Triton X-100; 10% glycerin; 1 mM dithiothreitol; 1 mM Na3VO4; 1 mM AEBSF; 60 μg/mL aprotinin; 0 μg/mL leupeptin; 1 μg/mL pepstatin A. The BC powder in buffer 1 (1:6 *v/v*, tissue/buffer) was incubated on ice for 5 min. The supernatant was separated by centrifugation (5 min, 1000× *g*, +4 °C). The pellet was lysed with buffer 2 (1:6 *v/v*, tissue/buffer) for 30 min at +2 °C. After centrifugation (30 min, 14,000× *g*, +4 °C), the supernatant containing the studied proteins was withdrawn, mixed with the loading buffer (4× Laemmli Sample Buffer), incubated for 5 min at 95 °C and stored at −80 °C. Protein concentration was measured spectrophotometrically according to the Bradford assay.

### 5.5. Measurement of CC Contents of Mitochondrial Enzyme Complex (MC) Subunits

The contents MC catalytic subunits measured in CC extracts using Western blot analysis with protein-specific antibodies [93]. The contents of the MC-I subunit, NDUFV2 (NADH dehydrogenase [ubiquinone] flavoprotein; MC-II subunit, SDHA (flavochrome subunit A of succinate dehydrogenase); MC-III subunit, Cyt b (cytochrome b); MC-IV subunit, COX1(cytochrome c oxidase subunit; ATP5A (ATP synthase alpha chain) were measured in the BC cytoplasmic extract Proteins from the samples were separated in polyacryamide gels (PAAG) (10%). The proteins were transferred from PAAG to a nitrocellulose membrane by electro elution for 60 min. The Western blots were preincubated (blockade of nonspecific antibody binding) in PBS containing 0.5% Tween-20 and low-fat 5% milk. Then, the Western blots were incubated in a 1:500 dilution of primary polyclonal antibodies (Santa Cruz Biotechnology, Dallas, TX, USA) for 14 h at +4 °C. After washout, the blots were incubated in a 1:5000 dilution of secondary antibodies conjugated with horseradish peroxidase (Santa Cruz Biotechnology, USA) for 60 min. The proteins were detected using the reaction with ECL reagents (Pierce Biotechnology, Inc., Pittsburgh, PA, USA) on a Kodak film and measured by densitometry using the Adobe Photoshop software. The protein content was estimated by optical density of the band reflecting the antibody binding to the protein. The result was expressed as relative densitometric units (RDU).

### 5.6. Measurement of Succinate Dehydrogenase Activity (SDH) in CC Mitochondria

Mitochondria were isolated from the CC tissue by differential centrifugation. SDH activity was determined spectrophotometrically [94]. The SDH activity was expressed as the amount of oxidized substrate in nmol per 1 mg of mitochondrial protein per min.

### 5.7. Electron Microscopy of the Cerebral Cortex

After decapitation of rats, the isolated fragments of the cortex were immediately fixed in 2.5% glutaraldehyde in 0.1 M cacodylate buffer (pH 7.4) for 2 h and then additionally fixed in 2% osmic acid prepared in the same buffer as described by Weakley [95]. The preparations were dehydrated in alcohol and acetone at increasing concentrations and embedded in Epon-812. At least five electron microscopic grids with ultrathin sections were prepared from each Epon block. The Epon blocks were cut into ultrathin sections (60–70 nm) using an ultramicrotome Leica (Vienna, Austria), stained with uranyl acetate and lead citrate and examined in the electron microscope (JEM 100-B, Tokyo, Japan).

Mitochondria were counted in the BC of LR and HR rats exposed to normoxic (control) and various hypoxic conditions on standard electron microscopic negatives (6 × 9 cm) at the same magnification. The number and size of mitochondria were determined and expressed as the percentage of the total number of mitochondria examined. The number of small mitochondria (0.14–0.25 μm) and elongated mitochondria (with the length more than twofold greater than the width) was counted and expressed as the percentage of the total number of mitochondria examined. At least 50 negatives for one experimental group were analyzed. The perimeter and area of mitochondria were determined using the Image J program (designed at the National Institutes of Health, Bethesda, MD, USA).

### 5.8. Statistics

Data were analyzed using the Statistica 6.0 software by the Student’s *t*-test and the Wilcoxon nonparametric rank U-test (Wilcoxon-Mann-Whitney). The difference between groups was considered significant at *p* < 0.05. Morphometrical data were analyzed using the Prizm for Windows (version 5.0) software. The factorial ANOVA test was followed by multiple comparison with the Bonferroni adjustment (Bonferroni’s Multiple Comparison Test). The difference was considered significant at *p* < 0.05.

## Figures and Tables

**Figure 1 ijms-22-08636-f001:**
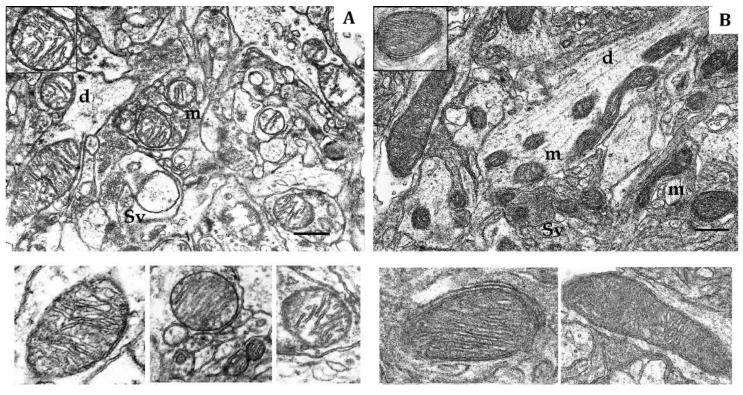
Electron micrography of rat cerebral cortex in normoxic conditions (21% O_2_). (**A**) rats with low-resistance to hypoxia; (**B**) rats with high-resistance to hypoxia. d, dendrite; m, mitochondria; Sv, synaptic vesicles. Scale bar: 0.5 μm. The bottom panels show representative mitochondria.

**Figure 2 ijms-22-08636-f002:**
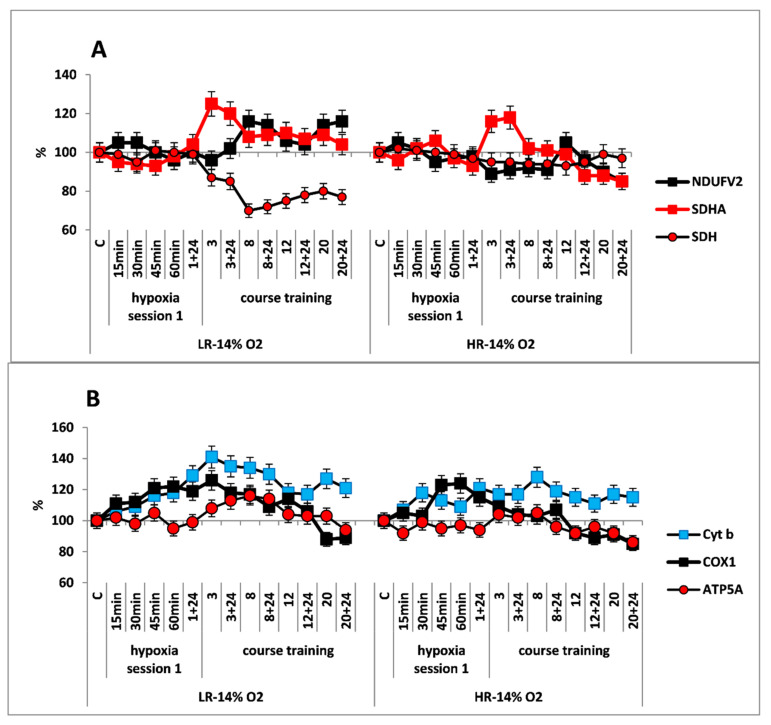
Effect of long-term hypobaric hypoxia (HBH, 14% O_2_; 20 daily one-hour sessions) on the dynamics of the CC content of mitochondrial enzymes in low-resistance (LR) and high-resistance (HR) rats. Abscissa: time points of the long-term hypoxic training, at which study parameters were measured: 15, 30, 45, 60 min: first hypoxic exposure; 3, 8, 12, and 20: number of conducted sessions; 3 + 24, 8 + 24, 12 + 24, 20 + 24: at 24 h after the latest hypoxic exposure. (**A**) NDUFV2 (MC I), SDHA (MC II) catalytic subunits (% of control) and SDH activity. (**B**) Cyt b (MC III), COX1 (MC IV) and ATP5A (MC V) catalytic subunits (% of control).

**Figure 3 ijms-22-08636-f003:**
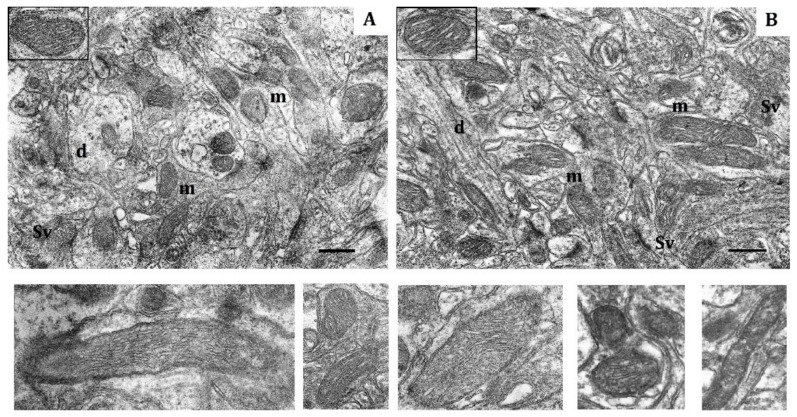
Effects of long-term hypobaric hypoxia (HBH-14% O_2_) on the rat cerebral cortex in electron micrography. (**A**) rats with low-resistance to hypoxia; (**B**) rats with high-resistance to hypoxia. Scale bar: 0.5 μm. For other designations, see legends to Figure 1.

**Figure 4 ijms-22-08636-f004:**
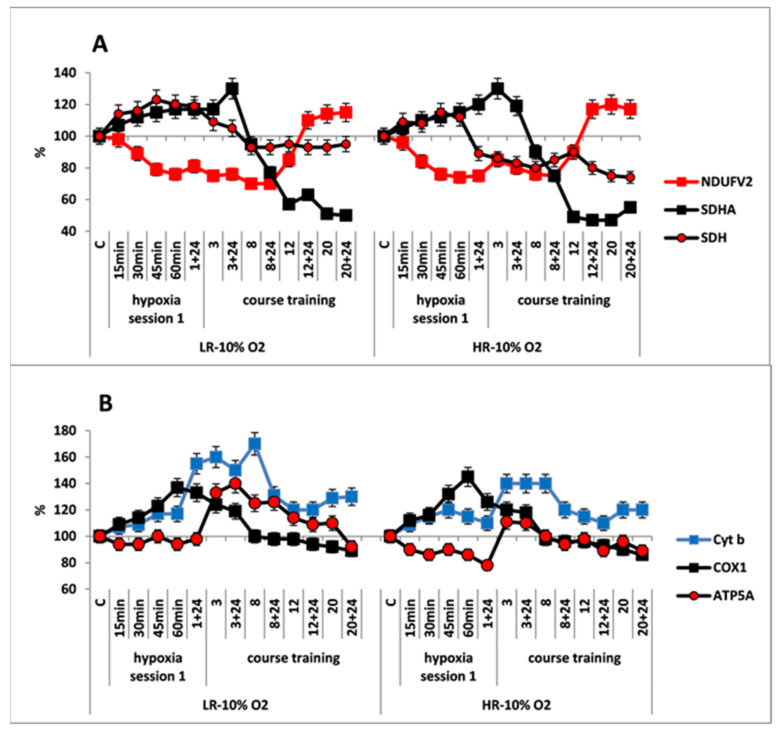
Effect of long-term hypobaric hypoxia (HBH, 10% O_2_; 20 daily one-hour sessions) on the dynamics of the content of mitochondrial enzymes in rats with low-resistance (LR) and high-resistance (HR) to hypoxia. (**A**) NDUFV2 (MC I), SDHA (MC II) catalytic subunits (% of control) and SDH activity. (**B**) Cyt b (MC III), COX1 (MC IV) and ATP5A (MC V) catalytic subunits (% of control). For other designations, see legends to Figure 2.

**Figure 5 ijms-22-08636-f005:**
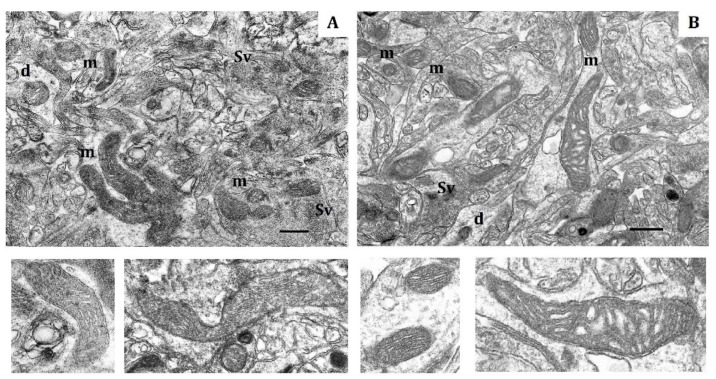
Electron micrograph showing the effect of long-term hypobaric hypoxia (HBH-10% O_2_) on mitochondria of the rat cerebral cortex. (**A**) rats with low resistance to hypoxia; (**B**) rats with high-resistance to hypoxia. Scale bar: 0.5 μm. For other designations, see legends to Figure 1.

**Figure 6 ijms-22-08636-f006:**
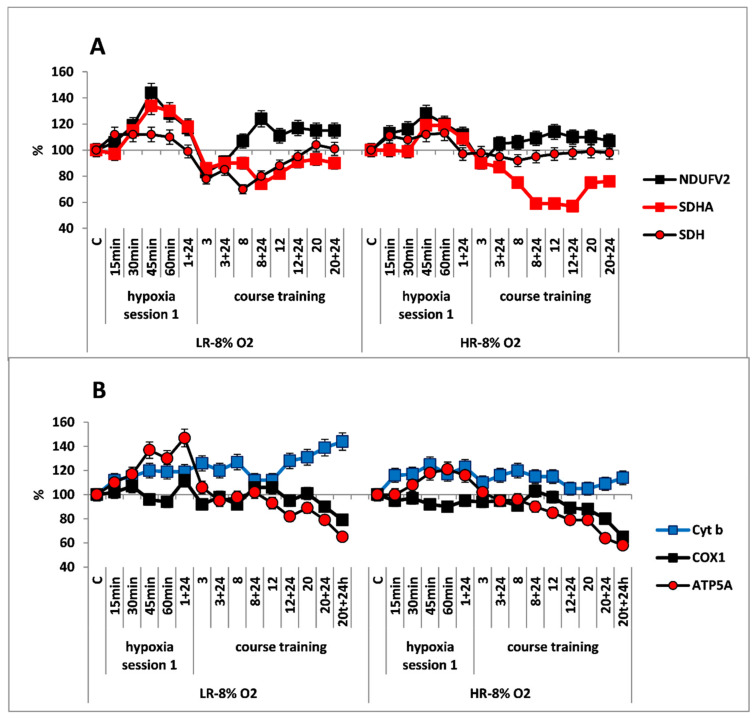
Effect of long-term hypobaric hypoxia (HBH, 8% O_2_; 20 daily one-hour sessions) on the dynamics of the CC content of mitochondrial enzymes in low-resistance (LR) and high-resistance (HR) rats. (**A**) NDUFV2 (MC I), SDHA (MC II) catalytic subunits (% of control) and SDH activity. (**B**) Cyt b (MC III), COX1 (MC IV) and ATP5A (MC V) catalytic subunits (% of control). For other designations, see legends to Figure 2.

**Figure 7 ijms-22-08636-f007:**
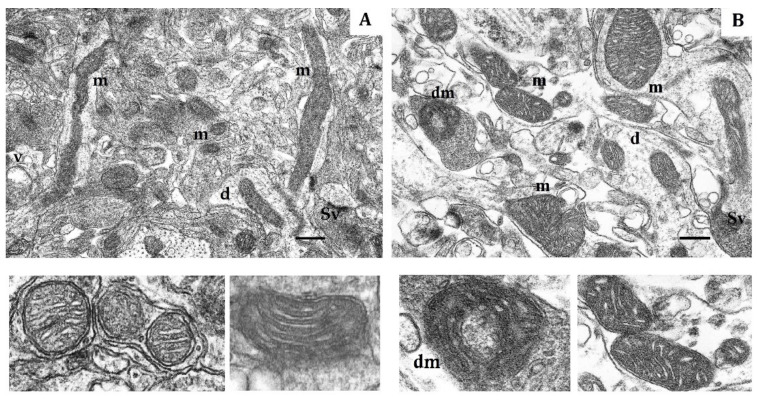
Electron micrograph showing the effects of long-term hypobaric hypoxia (HBH-8% O_2_) on mitochondria of the rat cerebral cortex. (**A**) rats with low-resistance to hypoxia; (**B**) rats with high-resistance to hypoxia. Scale bar: 0.5 μm. For other designations, see legends to Figure 1.

**Figure 8 ijms-22-08636-f008:**
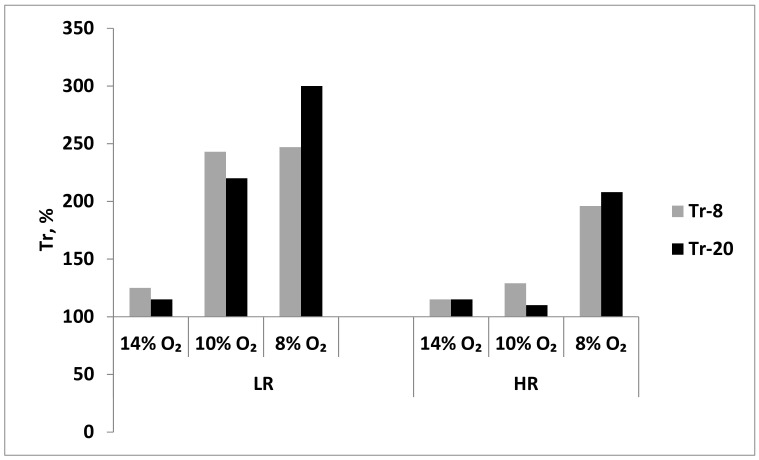
Effect of long-term adaptation (8 and 20 sessions) to various hypoxic regimens on Tr (resistance index, time to development of pathological breathing at a subcritical altitude in HBH).

**Table 1 ijms-22-08636-t001:** Contents of mitochondrial enzyme catalytic subunits (MC I-IV) in cerebral cortex (CC) of high-resistance (HR) and low-resistance (LR) rats (in % of control) in normoxia and following 20 sessions of one-hour hypoxic exposure in three different regimens.

Measured Parameters	Animal Phenotype	NDUFV2 (MC I)	SDHA (MC II)	Cyt b (MC III)	COX1 (MC IV)	ATP5A (MC V)
Control (21% O_2_)	HR/LR, %	155	125	129	119	119
**14% O_2_**						
14% O_2_ after 20 sessions	LR/C, %	116	104	121 *	89	94
	HR/C, %	85	85	115	85	86
	HR/LR, %	73	82	95	96	91
**10% O_2_**						
10% O_2_ after 20 sessions	LR/C, %	115 *	50 *	130 *	90	92
	HR/C, %	117 *	55 *	120	86	89
	HR/LR, %	102	110	92	96	97
**8% O_2_**						
8% O_2_ after 20 sessions	LR/C, %	115	90	145	79 *	65 *
	HR/C, %	107	76 *	114	65 *	58 *
	HR/LR, %	93	84	79 *	82	89

*—*p* < 0.05.

**Table 2 ijms-22-08636-t002:** Differences in morphometric values of mitochondria from cerebral cortex (CC) of high-resistance (HR) and low-resistance (LR) rats in normoxia.

Animal Phenotype	Total Number of Mitochondria per 10 µm^2^	Small Mitochondria (0.14–0.25 µm) (% of Total Number)	Elongated Mitochondria (0.5–3 µm) (% of Total Number)	Hyper-Elongated Mitochondria (4 µm) (% of Total Number)
HR	7.25 ± 0.43	32.8	38	11
LR	6.17 ± 0.36	28.6	29	3
HR/LR (%)	**118**	**115**	**131 ***	**367 ***

*—*p* < 0.05.

**Table 3 ijms-22-08636-t003:** Area and perimeter of small and elongated mitochondria in CC of HR and LR rats in normoxia.

Animal Phenotype	Area, µm^2^		Perimeter, µm	
	Small Mitochondria (0.14–0.25 µm)	Elongated Mitochondria (0.5–3 µm)	Small Mitochondria (0.14–0.25 µm)	Elongated Mitochondria (0.5–3 µm)
HR	0.047 ± 0.01	0.196 ± 0.014	0.751 ± 0.015	1.741 ± 0.111
LR	0.064 ± 0.03	0.281 ± 0.018	0.950 ± 0.022	2.030 ± 0.138
HR/LR (%)	73 *	70 *	79 *	86

*—*p* < 0.05.

**Table 4 ijms-22-08636-t004:** Effects of different hypoxic regimens on the shape and size of CC mitochondria in LR and HR rats.

Hypoxic Regimens	LR			HR		
	Small Mitochondria (0.14–0.25 µm) (in % of Control)	Elongated Mitochondria (0.5–3 µm) (in % of Control)	Hyper-Elongated Mitochondria (4 µm and >) (in % of Control)	Small Mitochondria (0.14–0.25 µm) (in % of Control)	Elongated Mitochondria (0.5–3 µm) (in % of Control)	Hyper-Elongated Mitochondria (4 µm and >) (in % of Control)
Immediately after the first hypoxic exposure						
**14% O_2_**	279 ± 3.0 **	92 ± 0.8	134 ± 1.1 *	124 ± 0.8	69 ± 0.5 *	69 ± 0.7 *
**10% O_2_**	357 ± 3.3 **	85 ± 0.6 *	85 ± 0.7 *	114 ± 1.0	78 ± 0.6 *	42 ± 0.3 *
**8% O_2_**	178 ± 1.9 **	138 ± 1.2 *	206 ± 1.5 **	47 ± 0.3 **	197 ± 2.0 **	438 ± 3.5 **
After course training						
**14% O_2_**	191 ± 1.5 **	134 ± 1.2 *	100 ± 0.9	105 ± 0.9	108 ± 1.1	64 ± 0.5 *
**10% O_2_**	199 ± 1.4 **	126 ± 1.1 *	300 ± 2.5 **	136 ± 1.1 *	95 ± 1.0	45 ± 0.4 **
**8% O_2_**	124 ± 1.1 *	98 ± 0.8	400 ± 3.8 **	80 ± 0.6 *	113 ± 0.9	45 ± 0.5 **

*—*p* < 0.05; **—*p* < 0.001.

## Data Availability

The data presented in this study are available on request from the corresponding authors.

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
