# Peer review of "Signaling Role of Mitochondrial Enzymes and Ultrastructure in the Formation of Molecular Mechanisms of Adaptation to Hypoxia"

_ijms, 2021, doi:10.3390/ijms22168636_

Round 1

Reviewer 1 Report

The topic by itself is interesting and relevant

With all respect, this reviewer finds the manuscript too long, unnecessarily "complex"

The presentation of the Results is particularly confusing and difficult to read/understand/follow

Considering the amount of works dedicated to the topic, including supercomplexes and their modulation under hypoxia, the experimental determinations may seem a little "weak" (of note, supercomplexes are addressed only in the discussion and should have been  evoked straight in the introduction  --but / and of course adressed/measured in the experimental section).

The selection/choice of the 5 complexes (of note, cyt c is not a complex) should be better rationalized

There are many many places in the ms with font changes/formating etc

The english language could certainly be improved (syntax , sentences etc)

Author Response

Dear Reviewer 1,

Thank you very much for the thorough review of the manuscript.

  1. With all respect, this reviewer finds the manuscript too long, unnecessarily "complex". The presentation of the Results is particularly confusing and difficult to read / understand / follow. The english language could certainly be improved (syntax, sentences etc)

We took into account your comments on the structure of the manuscript and did our best to simplify and make the text more readable. We also clarified some of the methodological details of our experiments and also took into account your comments concerning the English translation and thoroughly checked it.

2.There are many many places in the ms with font changes / formating etc

The copy of the work submitted by the editors contains typos (change of fonts, etc.). They are associated with the formation of an editorial copy. We will definitely inform the editor about this.

  1. Considering the amount of works dedicated to the topic, including supercomplexes and their modulation under hypoxia, the experimental determinations may seem a little "weak" (of note, supercomplexes are addressed only in the discussion and should have been evoked straight in the introduction -but / and of course adressed / measured in the experimental section).

There are indeed a huge number of works devoted to the molecular mechanisms of hypoxia. However, their findings are extremely contradictory. In addition, all of them were carried out on model cell systems, in vitro, at a very low O2 content (1 - 1,5%).

The aim of our work was to investigate in the mammalian brain the peculiarities of the reaction of the respiratory  chain substrate and cytochrome site enzymes and mitochondrial dynamics under various hypoxic effects in vivo. This is the first time such a multivariate and complex study is being carried out. For the first time, it was shown that all selected parameters are highly sensitive molecular markers of the energy metabolism state in a wide range of low oxygen concentrations. It is of great importance for understanding the role of mitochondria in cellular signaling.

The research  plans did not include the study of the supercomplexes formation in this process. This is the challenge for the future. However, in the Discussion, we considered the prospects and importance of this research.

  1. The selection / choice of the 5 complexes (of note, cyt c is not a complex) should be better rationalized

You write that the experimental part of the work is weak. However, this is for the first time that a comprehensive assessment of the interaction of enzymes of the substrate and cytochrome regions of the respiratory chain in combination with structural and morphological changes in mitochondria during hypoxia was carried out. For this, the subunits of mitochondrial complexes (MC) I-V (NDUFV2, SDHA, Cyt b, COX1, and ATP5A) were selected. (We did not measure Cyt c about which you write). Previously received our data on the OXPHOS reaction, the kinetic parameters of NADH-cytochrome c reductase and COX, as well as the ATP content under moderate hypoxia in the brain cortex, allowed us to understand the mechanism of interaction between MC II  and MCI. The data obtained in this work allowed us to draw three new conclusions that are important for mitochondriology:

  1. In the in vivo conditions, gradual changes in reduced oxygen content result in CC rats a differentiated response of MC I-IV enzyme subunits and in mitochondria ultrastructural changes. Together these parameters play a role of highly sensitive molecular markers reflecting the sate of energy metabolism and the formation of mechanisms of immediate and long-term adaptation of the mitochondrial apparatus to oxygen deficit.
  2. The МС II activation in a certain range of reduced O2 concentrations is a compensatory mechanism, that ensures the preservation of the activity of RC cytochrome site  enzymes, the formation of new kinetic properties of MCI and the restoration of its activity under these conditions, as well as the ultrastructural reorganization of mitochondria.
  3. The level of mitochondrial enzymes and mitochondrial dynamics in the CC LR and HR rats are different and affect the formation of urgent and long-term adaptation of animals to hypoxia.

Reviewer 2 Report

The authors here describe how the different components of respiratory chain/ enzymes and the mitochondrial structures get altered in the rat cerebral cortex get altered as a result of hypoxia and in conclusion show that RC can act as a sensor for oxygen concentration. Although, the study looks interesting there are additional items that need to be done in order to make the story stronger.

Major comments:

  • The authors need to show the images or representative images of the western blots either in the main figures or in a supplementary section for all the parts where they have used it as a tool to quantify subunit function.
  • Introduction needs some more content and could benefit from discussing pathological conditions that might occur as a result of changes oxygen homeostasis changes.
  • For the images that show the mito ultrastructure it important to show control images under normal conditions. 
  • Authors should perform some assays showing mitochondrial function utilizing platforms such as seahorse or orboros.
  • Additional parameters that can be shown here also are changes to mitochondrial membrane potential using dyes such mitotracker.
  • rt-pcr based methods should be utilized to see the changes in biomarkers of mito-function.

Minor comments: 

  • There are a lot of typos through out the manuscript which needs to be taken care Eg. line 89, line 724 -80 instead of -800?
  • Fonts are inconsistent throughout the manuscript.
  • It would be good to check the grammar and spellings etc. 

Author Response

Dear Reviewer 2,

Thank you very much for your work in reviewing our article. We took into account your comments as much as possible and tried to fix a lot.

Major comments:

1.The authors need to show the images or representative images of the western blots either in the main figures or in a supplementary section for all the parts where they have used it as a tool to quantify subunit function.

In our work, we performed spectrophotometric analysis of blots. The method makes it possible to estimate the spot density with an accuracy of up to ±5%. Visual assessment of the density of blots allows you to determine the difference between them, at best, with an accuracy of 20-50%. Therefore, the figures in the article were built on the basis of quantitative data and did not clutter them up with blots. In modern works, the use of blots is constantly decreasing. However, if the reviewers insist, we will definitely enter this data. (about 200 points.)

  1. Introduction needs some more content and could benefit from discussing pathological conditions that might occur as a result of changes oxygen homeostasis changes.

We agree with the remark and the corresponding text was inserted into the Introduction.

  1. For the images that show the mito ultrastructure it important to show control images under normal conditions

Figure 1 in the article is an electronographic image of mitochondria of both phenotypes under normoxic conditions (control)

  1. Authors should perform some assays showing mitochondrial function utilizing platforms such as seahorse or orboros.

The aim of our work was to investigate in the mammalian brain the peculiarities of the reaction of the substrate and cytochrome site  enzymes of the respiratory chain and mitochondrial dynamics under various hypoxic effects in vivo. It has been shown for the first time that all these parameters are highly sensitive molecular markers of the state of energy metabolism under conditions of different concentrations of O2. This conclusion is of great importance for understanding the role of mitochondria in cellular signaling. The tasks of the work did not include the study of these issues on other models. Moreover, considering the large volume of the manuscript, conducting research at other objects is not justified.

We also did not plan to study the formation of supercomplexes in these processes. This is the challenge for the future. Therefore, we did not consider this issue in the introduction. However, in the Discussion, we have considered the importance and value of such studies.

  1. Additional parameters that can be shown here also are changes to mitochondrial membrane potential using dyes such mitotracker.

In the Discussion, we presented the results of our earlier work on the OXPHOS reaction, the kinetic parameters of NADH-cytochrome c reductase and COX, as well as the ATP content under moderate hypoxia in the brain cortex. These data make it possible to evaluate the work of the energy apparatus of the cell under these conditions and makes it unnecessary to measure the membrane potential.

6.rt-pcr based methods should be utilized to see the changes in biomarkers of mito-function.

Thank you. We will take into account and use your remark in our future work.

Minor comments:

Thanks, everything was corrected in the manuscript.

Round 2

Reviewer 2 Report

Although the author have made some changes to the article based on the previous suggestions, but I still feel some additional items are still necessary which include.

 1. Yes please include the blot data.

 2. It would also necessary to show some form of assay that validates the mitochondrial function, at least showing a level showing the changes in the levels of biomarkers, which are key to mito function.

Author Response

Dear Reviewer,

Thank you very much for reviewing our article. We paid attention to your comments and tried to correct as much as we could.

Our comments:

  1. Yes, please include the blot data.

Upon your advice, we added the main blot data in the attachment.

  1. It would be also necessary to show some form of assay that validates the mitochondrial function, at least showing a level showing the changes in the levels of biomarkers, which are key to mito function.

We believe that ОXPHOS are recognized indicators of the energy state of the cell. Our previous studies have shown changes in these and other parameters in the brain under the different hypoxic exposures. They are provided in the Discussion.

Round 3

Reviewer 2 Report

All the criteria's have more or less been met. Also, thanks for including the representative blots. The text needs some minor editing. Thanks!

Author Response

Dear Reviewer 2,

Thank you very much for carefully reviewing our article.

Upon your request we had checked the article once again and made some slight changes. We hope you will approve it now.

As for the English language, we have already said that the translated article was thoroughly checked for grammar and other mistakes by Prof. Downey (University of North Texas Health Science Center, Fort Worth, USA). He is one of the top reviewers for The American Journal of Cardiology, it was stated in the previous edition of the article. However, according to your request, the text was doublechecked by another professional translator who had made only slights corrections.

The edited text is available on the website.